# Comparing Differences between Two Groups of Adolescents Hospitalized for Self-Harming Behaviors with and without Personality Disorders

**DOI:** 10.3390/jcm11247263

**Published:** 2022-12-07

**Authors:** Ping Wang, Chao Li, Marcos Bella-Fernández, Marina Martin-Moratinos, Leticia Mallol Castaño, Pablo del Sol-Calderón, Mónica Díaz de Neira, Hilario Blasco-Fontecilla

**Affiliations:** 1Faculty of Medicine, Autonomous University of Madrid, 28029 Madrid, Spain; 2Department of Psychiatry, Puerta de Hierro University Hospital, 28222 Majadahonda, Spain; 3Department of Psychology, Comillas Pontifical University, 28015 Madrid, Spain; 4Center of Biomedical Network Research on Mental Health (CIBERSAM), 28029 Madrid, Spain; 5Korian, 75008 Paris, France

**Keywords:** self-harm, personality disorders, non-suicidal self-injury, suicidal behavior, suicide attempt, adolescent

## Abstract

Self-harm (non-suicidal self-injury (NSSI) and suicidal behavior (SB)) is frequent display during adolescence. Patients with personality disorders (PDs) frequently self-harm. However, few studies have focused on the role of PDs in self-harming adolescents. In this study, we collected 79 adolescents hospitalized due to self-harm (88.6% female; 78.5% Caucasian) and divided them into two groups, with or without a diagnosis of PD. The socio-demographic and psychological-clinical data were collected through a structured interview by clinicians. Univariate, subgroup, and multiple logistic regression analyses were performed. Univariate analysis showed that adolescents with a PD and self-harm had (1) an older age at hospitalization (*p* < 0.01); (2) experienced physical and sexual abuse (*p* = 0.05, and *p* < 0.01, respectively); (3) ADHD (*p* = 0.05); (4) a greater number of SA (*p* < 0.01); and (5) probability of being a major NSSI patient (>20 lifetime NSSI episodes) (*p* < 0.01). After multivariate stratified analysis, the results indicated that an older age, and particularly major NSSI status were predictors of PD diagnosis. Early identification and a better understanding of the characteristics of adolescent PDs can assist clinicians in intervening earlier and developing more rational treatment strategies to reduce the long-term effects of PDs.

## 1. Introduction

Self-harm includes non-suicidal self-injury (NSSI) and suicidal behavior (SB). Both behaviors are common causes of adolescent hospitalization in acute inpatient units. The prevalence of personality disorders (PDs) is high in patients displaying self-harm. For instance, the prevalence of NSSI ranges between 50% and 80% among patients with self-harm [1]. Self-harm peaks during adolescence and is often the result of the interaction of numerous complex factors [2]. Consequently, it is critical to recognize and identify the unique characteristics of PDs in the self-harming adolescent population. In this study, we explore the differences in clinical characteristics between the two groups by comparing adolescents with PDs and adolescents without PDs in patients hospitalized for self-harm as a first step towards the implementation of early intervention programs.

The Diagnostic and Statistical Manual of Mental Disorders, Fifth Edition (DSM-5) describes PDs as “a persistent pattern of internal experience and behavior that deviates markedly from the expectations of an individual’s culture, is generalized and flexible, has onset in adolescence or early adulthood, stabilizes over time, and causes distress or impairment” [3]. The assessment and diagnosis of PDs in adolescents remain controversial, although the DSM-5 also states that if a PDs diagnosis is considered in childhood or adolescence, symptoms must be evident for at least one year [3]. Adolescence is a unique developmental period characterized by increased emotional reactions and sensitivities, impulsive behavior, and cognitive immaturity [4]. Thus, PDs throughout adolescence are frequently misdiagnosed as temporary, age-related deviant behaviors [5], and professionals tend not to diagnose personality pathology in patients during this period [6]. However, recent studies have found that persistent traits of personality and behavior are stable in early and middle childhood [7,8]. In addition, the literature suggests that maladaptive behaviors associated with personality pathology in adulthood are closely related to behavioral and emotional problems exhibited during adolescence [9,10]. The impacts of PDs are known to be permanent, and untreated or inappropriately treated may lead to deterioration mental health problems, interpersonal difficulties, and academic failure [8,10], and even serious behavioral problems in these adolescents, such as NSSI, SB [11], delinquency, social dysfunction, and substance abuse [12,13]. These behaviors, once developed, may be more difficult to treat and change later in life, and their long-term effects can be devastating.

Although there are few articles comparing adolescent self-harmers with and without PDs, descriptive analysis of a group of adult self-harmers indicates that those with a borderline personality disorder (BPD) may engage in more severe and frequent NSSI, as well as exhibit more frequent suicidal ideation and suicide attempts (SA) [14,15]. Based on previous studies, our main objective in this study was to compare the differences between adolescent self-harmers with PDs and without PDs in the following areas: (a) age of onset, frequency, or various NSSI and SA behaviors; (b) childhood traumatic experiences, peer relationships, and academic difficulty issues; and (c) comorbid psychiatric diagnoses. Given that personality stability increases with age [16,17] and those personality traits are more likely to change during childhood and adolescence; therefore, ensuring a clear understanding of the role of PDs in adolescent self-harmers is critical to provide earlier and specific treatment assistance to these adolescents.

## 2. Methods

### 2.1. Sample

Adolescent inpatients in the Child Psychiatric Unit at Puerta de Hierro University Hospital (Majadahonda, Madrid, Spain), hospitalized between 25 February 2021 and 17 June 2022 by self-harm (either a SA or NSSI) were asked to participate in the study, 79 patients were enrolled after signed consent was obtained by them and their legal guardians. Of them, twenty two were diagnosed with PDs, and 57 did not.The sample age ranged from 12 to 17 years. Seventy patients in the sample were female, seven patients experienced just SA behavior, and twenty-one patients experienced only NSSI behavior. A range of sociodemographic and psycho-clinical features of the patients were collected through structured interviews with clinicians.All psychiatric diagnoses of patients were confirmed by clinicians in the hospital. The main standard used for diagnosis is the DSM-5. The local ethics committee approved the study (IRB No. 82/20, 23 February 2021).

### 2.2. Measures

A protocol including sociodemographics, and a series of questionnaires used to evaluate childhood traumatic experiences, exposure to school bullying, and risk factors associated with SB were included. The diagnosis of PDs included patients diagnosed with dysfunctional personality dysfunction at discharge, but not identified as a determined type or diagnosed with single or multiple PDs. In the current study, we only used four suicide attempt items (41–44) of the Self-Injurious Thoughts and Behaviors Questionnaire (SITBI). The SITBI test–retest reliability varies from 0.47 to 0.87. In addition, they demonstrate strong inter-rater reliability (with KS ranging from 0.9 to 1) [18]. We also used the Spanish version of the Paykel Suicide Scale (PSS) which consists of a 5-item questionnaire. It shows good psychometric properties (Cronbach’s alpha = 0.93), and an acceptable fit to a one-factor model (factor loadings ranging from 0.57 to 0.87 and 36.9% of explained variance) [19]. We added two questions about the age of onset of NSSI and SA, as well as two questions about the frequency of NSSI and SA, to improve the information on SB. The Unbearable Psychache Scale (UP3), a 3-item scale (score range: 0–15) that predicts changes in suicide ideation, including general psychological distress, hopelessness, perceived burden, and low sense of belonging, was used to assess psychological distress [20].

In addition, the Spanish version of the 28-item Child Trauma Questionnaire-Short Form (CTQ-SF) and the Adolescent Peer Relations Instrument-Victimization (APRI-V) were utilized to assess maltreatment and bullying [21,22]. The CTQ-SF assesses various prevalent forms of abuse, including emotional abuse, physical abuse, sexual abuse, emotional neglect, and physical neglect; each item uses a 5-point Likert-type scale (1–5 points) with responses ranging from “never” to “almost always”. Cronbach’s alpha coefficients of the CTQ-SF scale were 0.87 for emotional abuse, 0.89 for physical abuse, 0.94 for sexual abuse, 0.83 for emotional neglect, and 0.83 for physical neglect (0.66) [23]. The APRI-V is a self-report questionnaire that assesses three aspects of bullying victimization: physical, verbal, and social. It consists of 19 items on a Likert-type scale that measures the frequency of experiencing victimization (1 = never, 2 = sometimes, 3 = most of the time, 4 = many times, 5 = consistently). If “sometimes” or more frequently was selected, the item was coded as a 1; otherwise, it was coded as a 0. The total score for each subgroup was used in this study. The internal consistency reliability of the Spanish version was satisfactory (Cronbach’s alpha = 0.943) [22].

### 2.3. Data Analysis

A series of univariable analyses were used to compare the demographic and psycho-clinical characteristics of individuals with and without personality disorders (PDs vs. Control). Continuous variables were described through mean and standard deviations, and categorical variables were described through absolute frequencies and percentages, the Mann–Whitney U test or ANOVA was utilized for continuous variables and the chi-square or Fisher’s Exact test for categorical variables (see Table 1). SA was coded as a dichotomous variable to distinguish the presence or absence of SA. Regarding NSSI, we used a Non-Suicidal Self-Injury Major Repeater (NSSI-MR) construct as an alternative to the NSSI frequency, since accurate numbers regarding the number of NSSI is usually difficult to recall by adolescents. The NSSI-MR is a patient with more than 20 NSSI behaviors in their lifetime [24]. Furthermore, subgroup analyses were performed in patients with SA and NSSI-MR to compare the differences in the experience of abuse in the groups with and without PDs. Finally, we used severe logistic regression models with different combinations of the variables that were statistically (*p* < 0.05) significant or close to significance (*p* < 0.1) in the univariate analysis to find out the factor that could correlate with the PDs. All analyses were performed in R (version 4.2) [25].

## 3. Results

In Figure 1, a Venn diagram shows the relationship between patients with PDs and NSSI and SA. 51.9% (n = 41) of the patients had both NSSI-MR and SA, of which 46.3% (n = 19) were diagnosed with PDs, and 53.7% (n = 22) were not diagnosed (see Figure 1). The mean age of the sample was 14.7 ± 1.54 years, and the group with a PDs (n = 22) was about one year older than the group without PDs (Control, n = 57) (see Table 1). Eighty-eight percent (n = 70) of the patients in the sample were female, 9% (n = 7) did not have NSSI, and 27% (n = 21) did not have SA. The first NSSI behavior in patients with NSSI behavior occurred between 5 and 17 years of age, with a mean age of approximately 13 years (12.90 ± 1.92 and 12.69 ± 2.48 for the PDs and controls groups, respectively). NSSI-MR and SA were more prevalent in the PDs group compared to controls. 90.91% (n = 20) of patients in the PDs group had NSSI-MR, twice as many as in the controls (45.61%) (n = 26). The majority of patients (87.34%, n = 69) had no more than 5 SA; in the PDs group, the mean was 2.72, the median was 3, and the range was 0–15; in the controls, the mean was 2.14, the median was 1, and the range was 0–20.

The group with PDs had higher scores for physical abuse (*p* = 0.05) and sexual abuse (*p* = 0.004). Emotional and physical neglect were more prevalent in the control group, although these differences were not statistically significant. The prevalence of attention-deficit/hyperactivity disorder (ADHD) and conduct disorder (CD) was higher in self-harmers with PDs than in controls, but the differences in CD did not reach statistical significance (*p* = 0.07; see Table 1).

We did a subgroup analysis splitting the patients with both NSSI-MR and SA into those with a PD (n = 19) and those without a PD (n = 22). The difference in clinical characteristics is shown in Table 2. There were no significant differences between the two groups, and only the sexual abuse difference is very close to statistical significance (*p* = 0.055). In the CTQ-SF, most of the patients in both groups experienced some degree of emotional abuse and physical/emotional neglect. Nevertheless, physical and sexual abuse showed a positive skew; most of them had minimum abuse, but another part had a higher score (see Figure 2). Regarding sexual abuse, patients with PDs had a high risk of sexual abuse (OR = 4.37, *p* = 0.03).

Finally, multiple logistic regression models were used to assess factors that may impact the diagnosis of PDs. In all models, the prevalence of PDs increased with age (OR = 1.65) (see Table 3). NSSI-MR was the most important risk factor for PDs in all models (OR = 14.86). Sexual abuse increased the risk of PDs (OR = 2.77) but did not reach significance in these models. Adolescents diagnosed with CD were five times more likely to display a PD.

## 4. Discussion

The relationship between personality and self-harm behaviors in adolescents has been poorly studied [15,26]. The recent theory of personality pathology considers that the influence of personality on self-harm is similar in adolescents and adults [26]. In this context, our findings regarding the diagnosis of PDs in adolescent self-harmers is associated with older age, sexual abuse, ADHD diagnosis, and particularly, NSSI-MR status is particularly relevant. Indeed, NSSI-MR increased around 15 times the probability of being diagnosed with PDs.

According to the results of our sample analysis, the probability of being diagnosed with PDs during adolescence increases with age. Studies of children in the community samples have shown that the highest prevalence of symptoms of PDs takes place during early adolescence [27] and tends to decrease in adulthood [28]. However, due to the special features of adolescents, there are few studies on age and onset trends in clinical samples. There is no consensus among clinicians on the diagnosis of PDs in adolescents. Until the DSM-5, the diagnosis of PDs could only be made in adult patients. This led to an age bias among physicians in diagnosing PDs.

Regarding the relationship between childhood trauma and PDs, the emotional, physical, or sexual abuse scores were higher in self-harmers with PD. However, in all samples and subgroups analyzed, sexual abuse was a robust risk factor for being diagnosed with PDs. Sexual abuse is an important risk factor in early BPD and worsens BPD symptoms [29,30]. On the other hand, we did not find bullying victimization to be a risk for PDs, but a study reported that chronic peer bullying may increase the risk of PDs by more than 7-fold. Bullying is a culturally heterogeneous factor, our sample was from Spain and was mostly female, which has a low rate of bullying, especially among women [22,31].

ADHD was the only clinical diagnosis associated with PDs in univariate analysis. This result is in keeping with the results in a 2 million population-based study that concluded that ADHD increased the risk of BPD around 20 times, while the association was stronger with women [32]. In multivariate analysis, we found that ADHD lost significance, while CD showed a possible increase in the probability of PDs. Previous studies also agreed that CD in adolescents is a strong predictor of antisocial personality disorder (ASPD), and Filip et al. (2014) concluded that conduct disorder is a necessary prerequisite for ASPD [8]. Interestingly Storebø et al., in a review published in 2016, showed by a review of 18 prospective studies that ADHD with or CD was a strong predictor of later development of ASPD [33]. Unfortunately, our study did not analyze this subgroup because of the sample size.

However, the most relevant finding of the present study is the close relationship between NSSI-MR and PDs diagnosis. Indeed, only an older age and, particularly, NSSI-MR remained significant in logistic regression analyses. In other words, the MR of NSSI was closely associated with a PD diagnosis in our sample. This is consistent with the results obtained from other studies [15,26,34]. Many factors that contribute to NSSI are also risk factors for PDs, such as emotional dysregulation, interpersonal discord, childhood trauma, etc. [14]. We also found that adolescent self-harmers with PDs were at greater risk of having SA compared with self-harmers without PDs. Some studies have suggested that those with PDs who have NSSIs may have a particularly high risk for SB [1]. In any case, this association was lost in multivariate analyses.

Our study has several limitations. First, the small sample size reduced our ability to detect small differences between groups. Second, we had no information regarding specific PDs diagnoses (i.e., BPD, ASPD, etc.). However, the categorical distinction of PDs was abandoned in DSM-5. Third, we used patient interviews and self-reports to collect questions related to self-injury. Accordingly, our results may be subject to biases such as concealment, exaggeration, and memory bias. Fourth, our sample was gender-biased because most participants were female, which in turn makes our findings less generalizable and limits the identification of gender differences. However, our results are congruent with literature, as most adolescent self-harmers are usually females. Fifth, the risk factors leading to self-harm and PDs may overlap, and the extent to which they impact each of the two requires more research. Future studies need to increase the sample size and balance the gender differences. Adolescence is an important period for the formation of personality. A longitudinal design with regular follow-up and clinical interventions could help observe the evolution and reduce the negative impacts of PDs on adolescent life. Future research could explore the differences in more characteristics of self-harm (age of onset, methods, etc.) in adolescents with and without PDs.

## 5. Conclusions

An older age, and particularly the presence of NSSI-MR, is closely related to the diagnosis of PDs in a sample of adolescent self-harmers. However, given the tautological nature of our findings (what is first, PD or NSSI?) and the transversal design of our study, we cannot derive etiological connections between PD and NSSI.

## Figures and Tables

**Figure 1 jcm-11-07263-f001:**
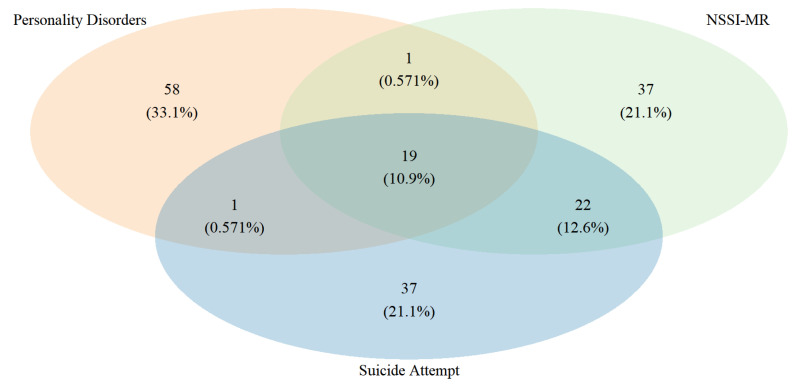
The Venn diagrams shown the logical relation between NSSI behavior, suicide attempts, and personality disorders in the sample (n = 79). NSSI-MR: Non-suicidal self-injury major repeater (NSSI > 20 times life span).

**Figure 2 jcm-11-07263-f002:**
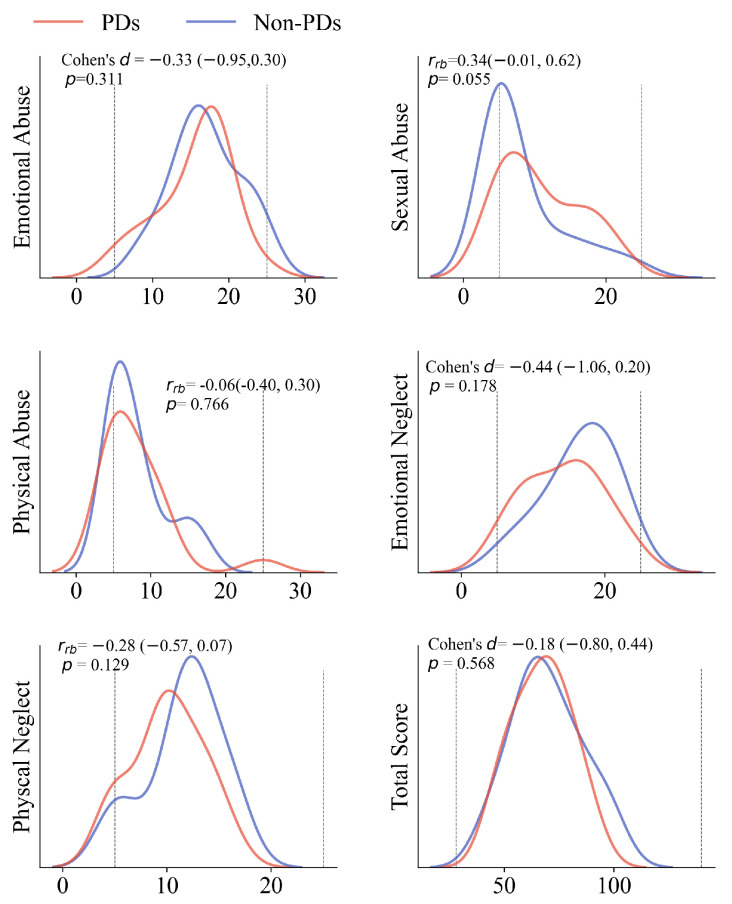
The kernel density estimation plots show the probability of the CTQ-SF scores in each group. Dashed lines show theoretical score ranges for each component. Effect sizes and *p*-values are reported in the plots.

**Table 1 jcm-11-07263-t001:** Sociodemographic and clinical-psychological features of the sample.

Variables	Total (n = 79)	PDs (n = 22)	Control (n = 57)	*p*-Value ^1^
**Age**	14.70 ± 1.54	15.27 ± 1.16	14.47 ± 1.62	**0.04**
**Female**	70 (88.6%)	19 (86.4%)	51 (89.5%)	0.70
**Ethnic**				0.54
Caucasian	62 (78.5%)	16 (72.7%)	46 (80.7%)	
Others	16 (21.5%)	6 (27.3%)	11 (19.3%)	
**Household Income**				0.39
<1500 Euro/month	15(18.99%)	5 (22.73%)	10 (17.50%)	
1500–2500 Euro/month	33 (41.78%)	11 (50.00%)	22 (38.60%)	
>2500 Euro/month	31 (39.23%)	6 (27.27%)	25 (43.90%)	
**Birth Order**				0.87
Only Child	8 (10.13%)	2 (9.10%)	6 (10.53%)	
Oldest Child	32 (40.50%)	10 (45.45%)	22 (38.60%)	
Others	39 (49.37%)	10 (45.45%)	29 (50.87%)	
**Screen Time**				
Social Media (hr.)	3.57 ± 2.38	4.32 ± 2.5	3.28 ± 2.29	0.12
Video Games (hr.)	0.58 ± 1.05	0.50 ± 0.80	0.62 ± 1.15	0.68
**Substance use**				
Alcohol	28 (35.44%)	9 (40.91%)	19 (33.33%)	0.71
Tabacco	23 (29.11%)	8 (36.36%)	15 (26.32%)	0.55
**NSSI Onset ^2^ (n = 72)**	12.75 ± 2.32	12.90 ± 1.92	12.69 ± 2.48	0.96
**SA Onset ^2^ (n = 58)**	13.87 ± 1.93	14.00 ± 1.89	13.80 ± 1.97	0.71
**NSSI-MR ^3^**	46 (58.23%)	20 (90.91%)	26 (45.61%)	**<0.01**
**SA**	2.72 ± 3.64	4.23 ± 4.38	2.14 ± 3.17	**<0.01**
**PSS ^4^**	4.63 ± 0.68	4.64 ± 0.90	4.63 ± 0.57	0.42
**UP3 ^4^**	11.97 ± 3.32	12.91 ± 3.1	11.55 ± 3.35	0.07
**SITBI ^4^**	10.27 ± 2.85	10.27 ± 2.21	10.27 ± 3.11	0.99
**CTQ-SF ^4^**	65.84 ± 13.90	69.45 ± 15.70	64.37 ± 12.97	0.15
Emotional Abuse	14.47 ± 5.30	15.73 ± 4.80	13.96 ± 5.45	0.19
Physical Abuse	7.38 ± 4.16	8.86 ± 5.54	6.78 ± 3.32	**0.05**
Sexual Abuse	8.30 ± 5.23	10.73 ± 5.79	7.31 ± 4.68	**<0.01**
Emotional Neglect	16.17 ± 5.61	15.14 ± 5.75	16.59 ± 5.54	0.31
Physcal Neglect	10.93 ± 3.38	10.50 ± 3.36	11.11 ± 3.41	0.48
**APRI-V** ^4^	4.37 ± 5.51	5.23 ± 5.90	4.02 ± 5.36	0.49
Social/Verbal Bullying	3.37 ± 4.02	4.09 ± 4.48	3.07 ± 3.82	0.61
Physical Bullying	1.00 ± 1.73	1.14 ± 1.70	0.94 ± 1.75	0.30
**Clinical Diagnosis**				
Major depressive disorder	30 (37.97%)	7 (31.8%)	23 (40.4%)	0.48
Anxiety disorders	6 (7.59%)	1 (4.5%)	5 (8.8%)	0.87
ADHD ^5^	23(29.11%)	10 (45.5%)	13 (22.8%)	**0.05**
Eating Disorders	13 (16.46%)	2 (9.1%)	11 (19.3%)	0.33
Substance Use Disorders	6 (7.59%)	2 (9.1%)	4 (7%)	0.67
Conduct Disorder	27 (34.18%)	11 (50%)	16 (28.1%)	0.07
Other Emotional Disorders ^5^	39(49.37%)	13 (59.1%)	26 (45.6%)	0.28

^1^ For continuous variables, the mean ± SD is reported and *p*-values are analyzed by ANOVA or Mann–Whitney U test, and for categorical variables, the number (n) and percentage (%) are reported and *p*-values are reported by chi-square test or Fisher’s exact test. ^2^ Age (year) of first occurrence of non-suicidal self-injury (NSSI) or Suicide Attempt (SA). ^3^ Non-suicidal self-injury major repeater (NSSI > 20 times life span). ^4^ PSS: The Paykel Suicide Scale; UP3: Unbearable Psychache Scale ; SITBI: The Self-Injurious Thoughts and Behaviors Questionnaire (items 41–44); CTQ-SF: The Childhood Trauma Questionnaire—Short Form; APRI-V: The Adolescent Peer Relations Instrument-victimization Spanish version. ^5^ ADHD: Attention-Deficit/Hyperactivity Disorder; Other Emotional Disorders: Emotional disorders with onset specific to childhood (ICD-10, F93).

**Table 2 jcm-11-07263-t002:** Comparison of features in subgroups self-harm of patients with and without personality disorders.

Variables	Total (n = 41)	PDs (n = 19)	Non-PDs (n = 22)	*p*-Value ^1^
**Age**	14.88 ± 1.54	15.32 ± 1.20	14.50 ± 1.71	0.11
**NSSI Onset ^2^**	12.32 ± 2.42	12.68 ± 1.86	12.00 ± 2.83	0.37
**SA Onset ^2^**	13.90 ± 1.97	14.11 ± 1.88	13.73 ± 2.07	0.55
**UP3 ^3^**	12.87 ± 2.41	12.30 ± 2.89	13.47 ± 1.65	0.17
**CTQ-SF ^3^**	68.63 ± 14.78	67.21 ± 13.45	69.90 ± 16.10	0.57
Emotional Abuse	16.32 ± 4.67	17.05 ± 4.52	15.53 ± 4.81	0.31
Physical Abuse	8.08 ± 4.32	8.05 ± 4.82	8.10 ± 3.92	0.77
Sexual Abuse	9.60 ± 5.81	10.95 ± 5.70	8.38 ± 5.77	0.06
Emotional Neglect	15.38 ± 5.38	16.48 ± 5.17	14.16 ± 5.48	0.18
Physcal Neglect	10.90 ± 3.56	10.11 ± 3.41	11.62 ± 3.61	0.13
**APRI-V ^3^**	4.37 ± 5.51	5.23 ± 5.90	4.02 ± 5.36	0.49
Social/Verbal Bullying	3.97 ± 4.29	3.84 ± 4.35	4.10 ± 4.35	0.62
Physical Bullying	1.00 ± 1.54	0.84 ± 1.17	1.14 ± 1.82	0.87
**Clinical Diagnosis**				
Major depressive disorder	18 (43.90%)	7 (36.84%)	11 (50.00%)	0.53
Anxiety disorders	2 (4.88%)	1 (5.26%)	1 (4.55%)	0.99
ADHD ^4^	14 (34.15%)	8 (42.10%)	6 (27.27%)	0.35
Eating Disorders	7 (17.07%)	2 (10.53%)	5 (22.73%)	0.42
Substance Use Disorders	4 (9.76%)	2 (10.53%)	2 (9.09%)	0.99
Conduct Disorder	14 (34.15%)	9 (47.37%)	5 (22.73%)	0.11
Other Emotional Disorders ^4^	21 (51.22%)	10 (52.63%)	11 (50.00%)	0.99

^1^ For continuous variables, the mean ± SD is reported and *p*-values are analyzed by ANOVA or Mann–Whitney U test, and for categorical variables, the number (n) and percentage (%) are reported and *p*-values are reported by Fisher’s exact test. ^2^ Age (year) of first occurrence of non-suicidal self-injury (NSSI) or Suicide Attempt (SA). ^3^ UP3: Unbearable Psychache Scale; CTQ-SF: The Childhood Trauma Questionnaire—Short Form; APRI-V: The Adolescent Peer Relations Instrument-victimization Spanish version. ^4^ ADHD: Attention-Deficit/Hyperactivity Disorder; Other Emotional Disorders: Emotional disorders with onset specific to childhood (ICD-10, F93).

**Table 3 jcm-11-07263-t003:** Multiple logistic regression analysis of potential factors influencing personality disorders and self-injury behaviors in adolescents.

	Age	NSSI-MR ^1^	Conduct Disorder	Sexual Abuse ^2^	ADHD ^3^	Physical Abuse ^4^	UP3 ^5^	Suicide Attempts	AIC
Cru.OR (95% CI)	1.45 (1.00–2.10)	11.3 (2.38–53.69)	2.27 (0.81–6.37)	4.03 (1.38–11.79)	2.88 (0.98–8.43)	1.89 (0.69–5.24)	1.16 (0.96–1.4)	4.00 (0.82–19.42)	
**Model 1**									77.30
*b* (SE)	0.56 (0.24) *	2.67 (0.95) **	1.25 (0.75)	0.82 (0.73)	0.86 (0.68)	−0.33 (0.69)	0.03 (0.13)	0.18 (1.10)	
Adj.OR (95% CI)	1.75 (1.08–2.83)	14.48 (2.24–93.67)	3.49 (0.80–15.21)	2.27 (0.55–9.48)	2.36 (0.61–9.10)	0.71 (0.18–2.77)	1.03 (0.79–1.35)	1.20 (0.14–10.36)	
**Model 2**									73.40
*b* (SE)	0.54 (0.23) *	2.74 (0.89) **	1.29 (0.75)	1.01 (0.68)	0.86 (0.67)	−0.37 (0.68)	-	-	
Adj.OR (95% CI)	1.72 (1.09–2.73)	15.43 (2.71–87.93)	3.65 (0.85–15.72)	2.75 (0.72–10.51)	2.36 (0.63–8.85)	0.69 (0.18–2.63)	-	-	
**Model 3**									71.63
*b* (SE)	0.52 (0.23) *	2.64 (0.86) **	1.29 (0.74)	0.97 (0.68)	0.80 (0.66)	-	-	-	
Adj.OR (95% CI)	1.69 (1.08–2.64)	13.94 (2.57–75.62)	3.62 (0.84–15.49)	2.63 (0.70–9.95)	2.23 (0.61-8.16)	-	-	-	
**Model 4**									70.99
*b* (SE)	0.51 (0.23) *	2.66 (0.85) **	1.32 (0.73)	1.02 (0.66)	-	-	-	-	
Adj.OR (95% CI)	1.67 (1.07–2.61)	14.33 (2.70–76.14)	3.75 (0.90–15.64)	2.77 (0.76–10.18)	-	-	-	-	
**Model 5**									70.85
*b* (SE)	0.50 (0.22) *	2.69(0.84) **	1.71 (0.69) *	-	-	-	-		
Adj.OR (95% CI)	1.65 (1.08–2.53)	14.84 (2.88–76.45)	5.55 (1.45–21.28)	-	-	-	-		

^1^ NSSI-MR: Non-suicidal self-injury major repeater, NSSI times > 20 in life span. ^2^ Sexual Abuse in CTQ-SF (score > 5). ^3^ ADHD: Attention-Deficit Hyperactivity Disorder. ^4^ Physical Abuse in CTQ-SF (score > 5). ^5^ UP3: Unbearable Psychache Scale. *: *p* < 0.05; **: *p* < 0.01.

## Data Availability

The corresponding author can provide the data described in this study upon request. Due to ethical and privacy constraints, the data are not publicly available.

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
