# Peer review of "Comparing Differences between Two Groups of Adolescents Hospitalized for Self-Harming Behaviors with and without Personality Disorders"

_jcm, 2022, doi:10.3390/jcm11247263_

Round 1
Reviewer 1 Report
This manuscript is presented in a well-structured manner and is interesting to the reader.
Several comments for the authors to consider -
2. Methods: It would be helpful to the reader to provide more details about the use of semi-structured clinical interviews. For example, who and/or where they were conducted and topic guides used. It is not clear what data analysis were used for the interviews or where the data is used within the study.
3. Results: please check that there is consistent use of 'P' or 'p', in regards to p-values.
4. Discussions: Suggest you highlight the strengths of the study alongside the limitations. Also it would be a nice addition if you added some thoughts with regards to future research.
Author Response
1. [Methods] It would be helpful to the reader to provide more details about the use of semi-structured clinical interviews. For example, who and/or where they were conducted and topic guides used. It is not clear what data analysis were used for the interviews or where the data is used within the study.
Response: We have incorporated your suggestions throughout the manuscript. In the subsection "Sample" we noted that "A range of sociodemographic and psycho-clinical features of the patients were collected through structured interviews with clinicians. All psychiatric diagnoses of patients were confirmed by clinicians in the hospital. The main standard used for diagnosis is the DSM-V. " (line 77-79)
2. [Results] please check that there is consistent use of 'P' or 'p', in regards to p-values.
Response: Thank you for pointing this out. We have corrected it.
3. [Discussions] Suggest you highlight the strengths of the study alongside the limitations. Also it would be a nice addition if you added some thoughts with regard to future research.
Response:We accept your suggestion, the opinions about future research have been added at the end of the "Discussion" section.
Reviewer 2 Report
TOPIC: Comparing differences between two groups of adolescents
hospitalized for self-harming behaviors with and without personality disorders
Thank you for inviting me to review this article. My comments are as follows:
TITLE
The title is satisfactory and represent the whole article
ABSTRACT
Abstract is well written consisting crucial elements where it provides the reader with the general view of this research. However, from my point of view the method used in this study is not clearly stated.
INTRODUCTION
The introduction is satisfactory
METHODS
1. Sample is satisfactory
2. It is good to combine the semi-structured interview with questionnaire and likert-type scale
3. P.3, line 107. Data analysis: The authors did not mention about the demographic characteristics of individuals in detail (i.e., Gender, Age etc.)
RESULTS
The discussion and interpretation of the results is good. The authors stated the demographic characteristics in describing result but it should be stated earlier in the methodology as well.
DISCUSSION
P.8.Line 159:” The relationship between personality and self-harm behaviors in adolescents has been poorly studied”. Do you have evidence to support your statement? Any research or citation?REFERENCES
Recent references should be added. This is because from 30 references, only 6 cited the most recent five-year reference
Author Response
1. [Abstract] However, from my point of view, the method used in this study is not clearly stated.
Response: Thank you for pointing this out. We have added methods and improved some English expressions in the abstract.
2. [Method] P.3, line 107. Data analysis: The authors did not mention about the demographic characteristics of individuals in detail (i.e., Gender, Age etc.)
Response:We have added a description in the subsection "Data Analysis" (lines 116–118); in the table's footnote, we also noted the information.
3. [Result]The authors stated the demographic characteristics in describing result but it should be stated earlier in the methodology as well.
Response: Agree, We added the demographic characteristics of the sample in the paragraph of subsection "Sample" and modified the paragraph with more details.
4. [Discussion]P.8.Line 159:" The relationship between personality and self-harm behaviors in adolescents has been poorly studied". Do you have evidence to support your statement? Any research or citation?
Response: We have added two references to support the statement (References 15 and 26).
5. [References]Recent references should be added. This is because from 30 references, only 6 cited the most recent five-year reference.
Response:As suggested by the reviewer, four references on recent research have been added to the paper (references 11, 12, 13, and 34).